# Genome-Wide Association Analysis of Boar Semen Traits Based on Computer-Assisted Semen Analysis and Flow Cytometry

**DOI:** 10.3390/ani15010026

**Published:** 2024-12-26

**Authors:** Xiyan Yang, Jingkun Nie, Yaxuan Zhang, Suqing Wang, Xiaoping Zhu, Zhili Li, Yunxiang Zhao, Xiuguo Shang

**Affiliations:** 1School of Life Science and Engineering, Foshan University, Foshan 528000, China; yangxy415@163.com (X.Y.); 1651405497@163.com (J.N.); ikokorange@163.com (Y.Z.); 15666932052@163.com (S.W.); zhuxiaoping@fosu.edu.cn (X.Z.);; 2School of Animal Science and Technology, Guangxi University, Nanning 530004, China; 3Guangxi Yangxiang Agricultural and Animal Husbandry Co., Ltd., Guigang 537100, China

**Keywords:** duroc boars, flow cytometry, GWAS, semen traits, CASA

## Abstract

Artificial insemination is crucial for pig breeding, underscoring the importance of semen quality in individual boars. This study explores the genetic basis of various semen traits, including MOT, DEN, ABN, MMP, AIR, and ROS levels, in Duroc boars. Semen traits were assessed by FCM and CASA. A total of 1183 Duroc boars were genotyped using the GeneSeek GGP Porcine 50 K SNP BeadChip. GWASs were conducted using GLM, MLM, and FarmCPU models. Heritability estimates for the traits were obtained using single- and multi-trait PBLUP models, which showed low heritability for all traits except ABN, which had medium heritability. Nine candidate genes, *GPX5*, *AWN*, *PSP-II*, *CCDC62*, *TMEM65*, *SLC8B1*, *TRPV4*, *UBE3B*, and *SIRT5,* were potentially linked to semen traits associated with antioxidant and mitochondrial functions in porcine sperm. These findings offer valuable insights into the genetic factors influencing semen quality and may improve economic outcomes in pig breeding through enhanced sperm quality and targeted breeding strategies. Incorporating these identified markers and genes into the genomic prediction models has the potential to improve the accuracy of genomic prediction for semen traits in Duroc pigs.

## 1. Introduction

The pig breeding industry is crucial for national food security, and the quality of breeding boars is fundamental to its success. With the rise in intensive and large-scale production, artificial insemination (AI) has become the primary method of commercial sow breeding. Semen quality significantly affects AI efficacy and the economic profitability of boar stations [1]. The decline in semen quality is one of the main factors leading to a shortened lifespan of boars, as boars with poor semen quality will be eliminated [2]. Currently, boars are mainly selected based on growth [3,4], carcass [5], and reproductive traits [6], such as litter size, with less emphasis on the genetic parameters of semen quality [7,8,9,10]. In recent years, there has been growing interest in understanding the molecular processes and genetic mechanisms that influence semen traits. Researchers integration of cross-descent meta-GWAS analyses, as well as discovery group transcriptome-wide association studies and colonic localization analyses of 524,426 semen quality trait records collected from 3248 individuals in five herds of pigs, revealed 176 non-overlapping key candidate genes associated with semen traits [11]. The researchers conducted a GWAS analysis on the semen characteristics of Duroc, Landrace, Large White [12], and Pietrain pigs [13], identifying several candidate genes. However, this analysis primarily focused on the conventional assessment of various semen quality parameters, including sperm concentration, motility, and morphology.

Incorporating semen traits into the boar selection criteria could enhance breeding programs and promote the balanced development of both production and reproductive performance. Semen traits, including sperm motility (MOT), semen density (DEN), and sperm abnormalities (ABN), are critical indicators of fertilization potential [14]. Advances in semen testing have introduced monitoring additional measures such as mitochondrial membrane potential (MMP), sperm acrosomal integrity rate (AIR), and reactive oxygen species (ROS) levels to comprehensively assess semen quality and predict reproductive success [15]. Computer-assisted sperm analysis (CASA) is commonly used to detect indicators such as MOT, DEN, and ABN in pig semen. In addition, CASA is also applicable to other species, such as dogs [16], bulls [17], and sheep [18], etc. MMP, AIR, and ROS need to be detected using flow cytometry in conjunction with fluorescent dyes.

Despite these advancements, the heritability of semen-related traits is low to moderate [19], which leads to slow genetic progress when using traditional breeding methods.

Recent developments in high-throughput genotyping and molecular technology have enabled the precise identification of quantitative trait loci (QTL) through genome-wide association studies (GWASs) [20]. GWASs have successfully mapped QTLs for key economic traits in both animal and plant breeding and identified genetic risk factors for human diseases [21]. Previous GWASs have identified candidate genes related to semen traits [13,22,23], such as *B9D2*, *PAFAH1B3*, *TMEM145*, and *CIC* for semen density, and *STRA8*, *ZSWIM7*, *TEKT3*, *UBB*, and *CATSPER1* for sperm motility [24,25].

However, many studies have focused on traditional semen indicators, with limited research on advanced metrics, such as MMP, AIR, and ROS levels. Candidate genes may vary across populations within the same breed, and mutations affecting spermatogenesis can affect semen quality. Therefore, the genetic markers and candidate genes related to semen traits should be further explored to improve our understanding of their genetic basis. The objective of this study is to utilize GWAS technology to identify QTL regions and candidate genes associated with semen quality in the Duroc pig population. The identified SNPs and candidate genes will enhance our understanding of the genetic mechanisms underlying semen traits and improve future breeding strategies in the pig industry.

## 2. Material and Methods

All animal care and experimental procedures used in this study met the guidelines of the Animal Care and Use Committee of Foshan University.

### 2.1. Animals and Phenotypes

Between January 2022 and July 2022, semen samples were collected from 1182 purebred Duroc boars at two boar stations operated by Guangxi Yangxiang Co., Ltd, Guigang China. Studies have shown that semen density is highest from January to March, sperm motility is highest from April to June [26], and January to July may be suitable for semen quality assessment. The pig barn has a solid concrete floor, with an automatic environmental control system, a positive pressure air filtration system, and a wet curtain fan to regulate the environment of the pig house, and the overall temperature is maintained at 18~22 °C. The boars were managed and fed according to the company standards, selected for breeding at 6 months of age, and conditioned for sperm collection starting at 7 months of age. The semen traits were assessed using CASA and flow cytometry (FCM). CASA was used to measure MOT, DEN, and ABN [27]. MMP was evaluated using Guava (HT 5.0) FCM with a JC-1 MMP assay kit and the results were expressed as the proportion of JC-1-positive cells [28]. The AIR was assessed by staining the sperm with propidium iodide (PI) and peanut agglutinin–fluorescein isothiocyanate (PNA–FICT), and the results were reported as the proportion of viable PI- and PNA–FICT-negative cells [29]. ROS levels were measured using Guava FCM and an ROS assay kit. The flow cytometer detected a cell count of 10,000, utilizing the green fluorescence channel for sample analysis, and the results were expressed as the proportion of DCFH-DA-positive cells [30].

Duroc boars range from 7–53 years of age, and outliers were removed based on the following ranges: 0.8–14 (×10^9^/mL) for DEN, 10–100% for MOT, and 0–100% for ABN, MMP, AIR, and ROS levels. Pedigree data spanning three generations included 7092 individuals after excluding incomplete or missing records.

### 2.2. Genotyping and Quality Control

Genomic DNA was extracted from semen samples using a magnetic bead-based kit (Wuhan NanoMagnetic Biological Company , Wuhan, China). Quality control criteria included clear agarose gel electrophoresis bands without tailing, total DNA concentration >1000 ng, and OD_260_/OD_280_ ratio between 1.6 and 1.8. Qualified DNA samples were genotyped using the GeneSeek Porcine 50 K SNP microarray (Nuqin Biotechnology Co., Ltd., Shanghai, China), which generated 50,696 SNPs. Quality control of the genotyping data was performed using Plink 1.9 software, with criteria including the removal of sex chromosomes and SNPs at unknown locations, exclusion of individuals with a genotype call rate below 90%, SNP call rate below 90%, minor allele frequency below 5%, and Hardy–Weinberg equilibrium *p*-value < 10^−6^. Missing data were imputed using the Beagle 4.1 software, and a second quality control step was conducted. Following these procedures, data from the remaining individuals were analyzed. Linkage disequilibrium (LD) analysis was performed using PopLD decay (https://github.com/BGI-shenzhen/PopLDdecay, accessed on 20 December 2023) using default parameters.

### 2.3. Estimation of Genetic Parameters

R(4.1) software was used to organize the collected semen data and trace the 3-generation pedigree of Duroc boars containing a total of 7092 individuals, excluding incomplete and missing pedigrees. After data quality control, 32,196 SNP loci and 1127 semen assays from Duroc boars remained. By analyzing the factors affecting semen traits using a general linear model, factors that did not have a significant effect on the model were eliminated, and those that were significant were added to the subsequent analyses, thus identifying each of the fixed effects as the source farm, semen harvesting season, and semen harvesting month of age. The field effect was divided into two levels based on the number of semen collection fields, the seasonal effect was divided into three levels based on the semen collection season (January in winter, March to May in spring, and June to July in summer), and the monthly effect was divided into five levels based on the age of the boar at the time of semen collection (7–12, 13–24, 25–36, 37–48, and 49–53). Narrow-sense heritability of semen traits in Duroc boars was estimated using the ASReml 4.0 software with the PBLUP model for both single and multiple traits. Genetic and phenotypic correlations among semen traits were calculated using the PBLUP model for multiple traits. Statistical modeling based on the PBLUP method is described as follows:Y=Xb+a+e

Here, Y represents the phenotypic values of the semen traits, ***b*** denotes the vector of fixed effects, **α** is the vector of additive effects, **e** represents the vector of random residual effects, and X is an incidence matrix. This model was used to estimate the genetic parameters of semen traits, and the specific formula was as follows:$$\mathrm{Heritability}\text{:}\;h^{2}=\frac{\sigma_{a}^{2}}{\left(\sigma_{a}^{2}+\sigma_{e}^{2}\right)}$$
$$\mathrm{Genetic}\;\mathrm{correlation}\text{:}\;r_{g}=\frac{Cov\left(a1,a2\right)}{\sqrt{\sigma_{a1}^{2}\sigma_{a2}^{2}}}$$
$$\mathrm{Phenotypic}\;\mathrm{correlation}\text{:}\;r_{p}=\frac{Cov\left(p1,p2\right)}{\sqrt{\sigma_{p1}^{2}+\sigma_{p2}^{2}}}$$

Here, h^2^ represents trait heritability; σ^2^_a_ and σ^2^_e_ denote additive genetic variance and residual variance, respectively. Cov(a1, a2) is the covariance between the additive genetic effects of two traits; σ^2^_a1_ and σ^2^_a2_ are the additive genetic variances for traits one and two, respectively. Cov(p1, p2) represents the phenotypic covariance between the two traits, while σ^2^_p1_ and σ^2^_p2_ are the phenotypic variances for traits one and two, respectively.

### 2.4. Identification of Semen Trait Genes Using GWAS

GWASs were conducted using R (Rx64 4.0.3, https://www.r-project.org/, accessed on 20 December 2023) software with three models: generalized linear model (GLM), mixed linear model (MLM), and fixed and random model circulating probability unification (FarmCPU) model. The MLM model is commonly used and described by the following formula:y=Wa+Zb+Sc+e

Here, y is a vector of phenotypic values of Duroc semen traits; **α** is the single nucleotide polymorphism substitution effect; **b** is a fixed effect including semen collection site, semen collection age in months, and semen collection season; **c** is a randomized additive effect obeying **c**~N(**0**,**G**$\sigma_{g}^{2}$), where **G** is the genomic relationship matrix constructed using single nucleotide polymorphism marker information and is the unknown variance component; W, Z, and S are incidence matrices for **α**, **b**, and **c**, respectively; and **e** is a vector of randomized residual effects.

The FarmCPU model iteratively uses fixed effects and random effects, and each genetic marker is tested using a fixed effects model. Potential quantitative trait nucleotides (QTNs) were added as covariates in the model for association analysis. The fixed effects model can be written as follows:$$Y_{i}=Xb+a_{1}\times PC_{1}+\cdots a_{3}\times PC_{3}+b_{1}\times G_{i1}+\cdots b_{i}\times G_{it}+d_{j}\times S_{ij}+\varepsilon_{i}$$

Here, Y_i_ is the semen trait of the *i*th individual; X is the corresponding incidence matrix; **b** is the fixed effect, including the top three principal component effects semen collection site, semen collection month, and semen collection season; **PC_1_**, **PC_2_**, and **PC_3_** are the top three principal components obtained from the genome-wide genotype data; and a_1_, a_2_, and a_3_ represent the corresponding effects. **G_it_** is the genotype of the *i*th potential associated with traits, and b_1_, b_2_, and b_i_ are the corresponding effects. The first iteration is empty. S_ij_ refers to the genotype of the *j*th genetic marker for the *i*th individual, and d_j_ is the corresponding effect. **ε_i_** is the residual effect vector with the distribution **ε**~(**0**, **I**$\sigma_{\varepsilon }^{2}$), where $\sigma_{\varepsilon }^{2}$ represents the residual variance. The random effects model uses the SUPER algorithm to optimize different combinations of possible associated QTNs using the *p*-value and genetic marker location information. The random effects model can be written as follows:$$Y_{i}=\mu_{i}+\varepsilon_{i}$$

Here, Y_i_ denotes the semen traits of individual boars; μ_i_ is the total genetic effects of the *i*th individual and defined by μ~N(0, ${K\sigma }_{u}^{2}$), where K is the kinship matrix defined by pseudo QTNs and $\sigma_{u}^{2}$ is an unknown genetic variance; **ε_i_** is the residual effect vector.

The GLM lacks a random effect term compared to the MLM, which can be described as follows:y=Wa+Zb+e

Here, y is a vector of phenotypic values of semen traits in Duroc; **α** is the single nucleotide polymorphism substitution effect; **b** is a vector of fixed effects; W and Z are the incidence matrices of α, b, and c, respectively; and **e** is the residual effect vector. In this study, Bonferroni correction was used to correct the *p*-value to −log_10_(*p*-value) = 4.49, *p*-value = 1/N, as the threshold of potential significance for genome-wide association of semen traits in the Duroc, where N represents the number of SNPs remaining after quality control.

### 2.5. Identification and Functional Annotation of Significant Loci

The Ensembl database was used to identify potential candidate genes within 1 Mb (upstream and downstream) of the genome-wide significant single nucleotide polymorphisms on the Sus scrofa 11.1 genome (http://asia.ensembl.org/, accessed on 20 March 2024). Candidate genes were associated with traits based on their biological functions.

## 3. Results

### 3.1. Phenotypic Variation and SNP Genotyping

Summary statistics for MOT, DEN, ABN, AIR, MMP, and ROS levels are presented in Table 1. Phenotypic distribution is shown in Appendix A, demonstrating that all the traits were normally distributed. The coefficients of variation ranged from 4.08% to 90.96%, with MOT and ROS levels exhibiting the smallest (4.08%) and largest (90.96%) coefficients, respectively. The factors contributing to the increase in ROS are multifaceted. In addition to heat stress, elements such as the types and quantities of microorganisms present in semen, non-standardized semen collection procedures, and inadequate detection methods may also play a significant role in this phenomenon. After quality control of the gene chip sequencing data, 1127 individuals and 32,196 SNPs were retained for further analysis. The SNP density plots for each chromosome are shown in Appendix A, and scatter plots for the first two principal components of the PCA are depicted in Appendix A.

### 3.2. Genetic Parameter Estimation

The estimates of the genetic parameters and their standard errors are provided in Table 2. Estimates of heritability for both DEN and MMP traits using the single-trait PBLUP model were lower than those from the multiple-trait model, but the difference was not significant, both differing by only 0.01. The multiple-trait model may yield higher estimates of heritability compared to the single-trait model by leveraging the genetic correlations among traits and more effectively decomposing additive and non-additive genetic effects. For the other four semen traits, the heritabilities estimated by the single-trait PBLUP model were all higher than those estimated by the multiple-trait model. Across all traits, heritability estimates were generally low, except for DEN, which showed a high heritability. ABN exhibited the lowest heritability (0.10).

Multi-trait models can make better use of phenotypic and genetic correlations of traits to assess individuals for two or more traits, and when both low and high heritability traits were analyzed together, the low heritability traits were more beneficial. Genetic covariance and residual covariance among traits are taken into account by multi-trait models, thereby increasing the accuracy of ratings. In this study, genetic and phenotypic correlations among semen traits were estimated using a multi-trait PBLUP model. Table 3 presents the genetic correlation coefficients: ROS was strongly negatively correlated with AIR (r = −0.57) and MMP (r = −0.73). DEN was strongly positively correlated with MOT (r = 0.86), and MMP was strongly positively correlated with AIR (r = 0.93). AIR was moderately positively correlated with MOT (r = 0.36), whereas other traits had weak genetic correlations. Phenotypic correlations were moderate between DEN and MOT (r = 0.32) and between MMP and AIR (r = 0.38), whereas other phenotypic correlations were weak.

### 3.3. Population Structure and Linkage Disequilibrium Decay

Population stratification can lead to false positive results in genetic studies [31]. To address this, the first three principal components were included as covariates in the GWAS model (Appendix A). Furthermore, linkage disequilibrium (LD) analysis was performed using pooled data, with the results showing that LD decay tended to be stable at distances of 1 Mb (Appendix A). Consequently, genes within 1 Mb of the significant SNPs were designated as candidate genes.

### 3.4. GWAS Results for Semen Traits

Appendix A and Figure 1 present the GWAS results for semen traits, including genomic positions, candidate genes, distances, and *p*-values. Seventy SNPs associated with semen traits were identified in the three GWAS models. Forty-three SNP loci associated with the DEN trait were identified, among which three are significant loci, forty are potential significant loci, and all are common loci identified by both FarmCPU and GLM models. The three significant loci, all located on chromosome 14, are WU_10.2_14_4278229, MARC0028432, and MARC0017786. Six SNP loci associated with the MOT trait have been identified, which are distributed across chromosomes 1, 7, and 16. These loci are designated as potentially significant and are recognized as common loci by both the FarmCPU and GLM models. Two SNP loci associated with ABN traits were found on chromosome 8, both of which were potentially significant and common to both the FarmCPU and GLM models. Twelve SNP loci associated with MMP traits were found on chromosomes 4 and 14, which were potentially significant loci. In addition to locus MARC0029602, which is common to the MLM, FarmCPU, and GLM models, the remaining 11 loci are common to the FarmCPU and GLM models. Five SNP loci associated with the AIR trait were found on chromosome 14, all of which were potentially significant and common to both the FarmCPU and GLM models. Two SNP loci associated with ROS traits were found on chromosomes 1 and 12, which were both potentially significant loci. The SNP locus WU_10.2_7_10722135, detected by the MLM model, had the highest *p*-value significance of 2.94E−05, and was located in the gene GFOD1, which is flanked by the genes RNF182 and CD83, and the other locus was common to both the FarmCPU and GLM models. Except for the three SNPs that were significantly associated with DEN, other SNPs were considered potentially significant loci. The MLM model identified only one potentially significant locus associated with the ROS trait. The FarmCPU, GLM, and MLM models all identified locus MARC0029602 as being associated with MMP. The remaining SNP loci associated with semen traits are common to both FarmCPU and GLM models.

### 3.5. Identification of Candidate Genes and Functional Enrichment Analysis

By reviewing previous studies on gene-related traits and using database search functions, nine potential candidate genes for semen traits in Duroc boars were identified within approximately 1 Mb upstream and downstream of each significant or potentially significant SNP. *GPX5* is a key candidate for MOT. *AWN*, *PSP-II*, and *CCDC62* were recognized as important candidates for AIR. *TMEM65*, *SLC8B1*, *TRPV4*, and *UBE3B* were identified as significant candidates for MMP. *SIRT5* has been highlighted as a major candidate for ROS levels.

These findings are crucial for understanding the genetic underpinnings of these traits and have significant implications for future research. *AWN*, *PSP-II*, and *CCDC62* are associated with the structure and function of sperm acrosomal membranes. *TMEM65*, *SLC8B1*, *TRPV4*, and *UBE3B* were involved in mitochondrial oxidative stress responses and calcium regulation, whereas *SIRT5* was involved in regulating antioxidant responses (Table 4).

## 4. Discussion

Semen traits are crucial quantitative indicators of pig production. However, many of these traits are complex and exhibit low heritability, making genetic improvement more challenging than other livestock traits. In this study, we estimated the heritability of semen traits in Duroc pigs using two animal models. For the more accurate multi-trait PBLUP model, the heritability estimates were 0.29 for DEN, 0.19 for MOT, 0.13 for ABN, 0.18 for MMP, 0.11 for AIR, and 0.14 for ROS levels. According to conventional heritability classifications, DEN has medium heritability (0.2 ≤ h^2^ < 0.4), while MOT, ABN, MMP, AIR, and ROS levels have low heritability traits (h^2^ < 0.2).

Yifeng et al. used a single-trait model to estimate the heritability of VOL, MOT, DEN, and ABN as 0.29, 0.10, 0.16, and 0.15, respectively [32]. Xiaoke et al. estimated DEN and ABN heritabilities as 0.28 and 0.298, respectively, using DMU V6 software with single- and multiple-trait models and found moderate heritability for all semen traits [33]. Li et al. employed a multi-trait model to analyze seasonal effects and estimate genetic parameters in southern China, with heritabilities of 0.42, 0.34, and 0.16 for MOT, DEN, and ABN in Duroc, respectively [13]. Smital et al. reported heritabilities of 0.58 for VOL, 0.34 for MOT, 0.38 for DEN, and 0.49 for AIR using a multi-trait model [34]. Marques et al. used the ASReml 3.0 multi-trait model to evaluate semen quality in various pig breeds and found heritabilities of 0.17 for MOT and 0.24 for ABN in the Duroc [35]. The heritabilities of DEN, MOT, and ABN in this study are consistent with those reported by Hong et al., although the estimates for MMP, AIR, and ROS levels are novel in the context of Duroc boars in China. MMP influences sperm motility by regulating mitochondrial function, and it has been found that the mitochondrial membrane potential of sperm is significantly positively correlated with their motility [36]. The acrosome is essential for sperm to undergo the acrosome reaction, penetrate the zona pellucida of the oocyte, and successfully participate in fertilization [37]. The production of ROS is a common characteristic of mature sperm cells and is a significant factor leading to oxidative stress [38]. Therefore, the estimation of heritability for MMP, AIR, and ROS provides a relevant data foundation and new directions for the breeding and improvement of breeding pigs. Variability in heritability estimates across studies may be attributed to differences in population genetic structures, sample sizes, management practices, and the statistical models used.

Genetic and phenotypic correlations are fundamental for understanding quantitative traits and guiding practical breeding. Phenotypic correlations reflect the relationship between two traits measured in the same individual, whereas genetic correlations reflect the association between breeding values and traits. These correlations are essential for breeding selection and can guide the indirect selection of traits that are difficult to measure. Genetic correlations were assessed using a multi-trait PBLUP model. ROS levels exhibited strong negative correlations with AIR (r = −0.57) and MMP (r = −0.73). DEN was strongly positively correlated with MOT (r = 0.86) and MMP with AIR (r = 0.93). AIR showed a moderately positive correlation with MOT (r = 0.36). Compared with other studies, such as those by Li et al., which found DEN and MOT to be strongly correlated (r = 0.304) [19,33,35,39], our results indicate some variation, likely due to different analytical models and breeding management practices. There is a positive correlation between DEN and MOT. In practical production, it is common to select semen with higher density to improve the success rate of breeding.

The advent of high-throughput SNP genotyping has made GWAS a powerful tool for identifying genetic markers associated with complex traits [40,41,42]. However, the choice of the GWAS model can impact the results. GLMs account only for fixed effects, whereas MLMs and FarmCPU models incorporate kinship matrices as random effects, potentially reducing false positives but sometimes increasing false negatives owing to overfitting [43,44]. The FarmCPU model is an iterative approach that improves computational speed and accuracy and enhances candidate gene identification [45,46,47]. In this study, GLM, MLM, and FarmCPU models were used to conduct a GWAS for semen traits in Duroc pigs. A total of 43 SNPs were associated with semen traits, with MLM identifying only 1 SNP each for MMP and ROS levels, indicating limited utility for these traits [43,46,48]. In contrast, the GLM and FarmCPU models identified more SNPs and candidate genes [46,47,49], particularly for traits with low heritability.

Based on these significant and potentially significant loci, we identified nine candidate genes for semen traits in Duroc boars. *GPX5* has emerged as a significant candidate for MOT because it encodes glutathione peroxidase 5, which is critical for maintaining sperm DNA integrity and mitigating oxidative stress [7,8,9,10,50]. *AWN*, *PSP-II*, and *CCDC62* have been identified as key candidates for AIR, with *AWN* and *PSP-II* playing roles in sperm adhesion and *CCDC62* being crucial for sperm motility [51,52,53,54,55,56,57]. For MMP, *TMEM65*, *SLC8B1*, *TRPV4*, and *UBE3B* have been highlighted, with *TMEM65* being involved in the mitochondrial oxidative stress response and *SLC8B1*, *TRPV4*, and *UBE3B* regulating mitochondrial calcium transport [58,59,60,61,62,63]. *SIRT5*, identified as a candidate for ROS levels, contributes to oxidative stress regulation and cellular homeostasis [64,65]. These findings offer valuable insights for future research and breeding programs aimed at improving the semen quality in Duroc pigs.

## 5. Conclusions

In this study, we employed three models, GLM, MLM, and FarmCPU model, for a GWAS of semen traits in 1183 purebred Duroc pigs. The analysis identified 43 SNP loci associated with semen density (DEN), 6 SNP loci with motility (MOT), 2 SNP loci with abnormality (ABN), 12 SNP loci with membrane integrity (MMP), 5 SNP loci with acrosome integrity (AIR), and 2 SNP loci with ROS levels. Notably, three significant SNPs were identified in DEN. Nine candidate genes associated with semen traits were identified: *GPX5*, *AWN*, *PSP-II*, *CCDC62*, *TMEM65*, *SLC8B1*, *TRPV4*, *UBE3B*, and *SIRT5*. These genes are involved in crucial functions, such as sperm antioxidation, acrosome membrane structure and function, mitochondrial oxidative stress response, and intracellular Ca^2+^ regulation. These findings have substantial implications for advancing marker-assisted genomic selection in pig breeding. In the future, existing research findings can be integrated with modern molecular biology techniques such as genome editing, yeast two-hybrid assays, and cDNA library construction to further explore the genes associated with boar semen traits. Additionally, through genetic modification, the quality of boar semen can be further improved, optimizing the genetic characteristics of the breed and providing a beneficial basis for subsequent genetic breeding research in boars.

## Figures and Tables

**Figure 1 animals-15-00026-f001:**
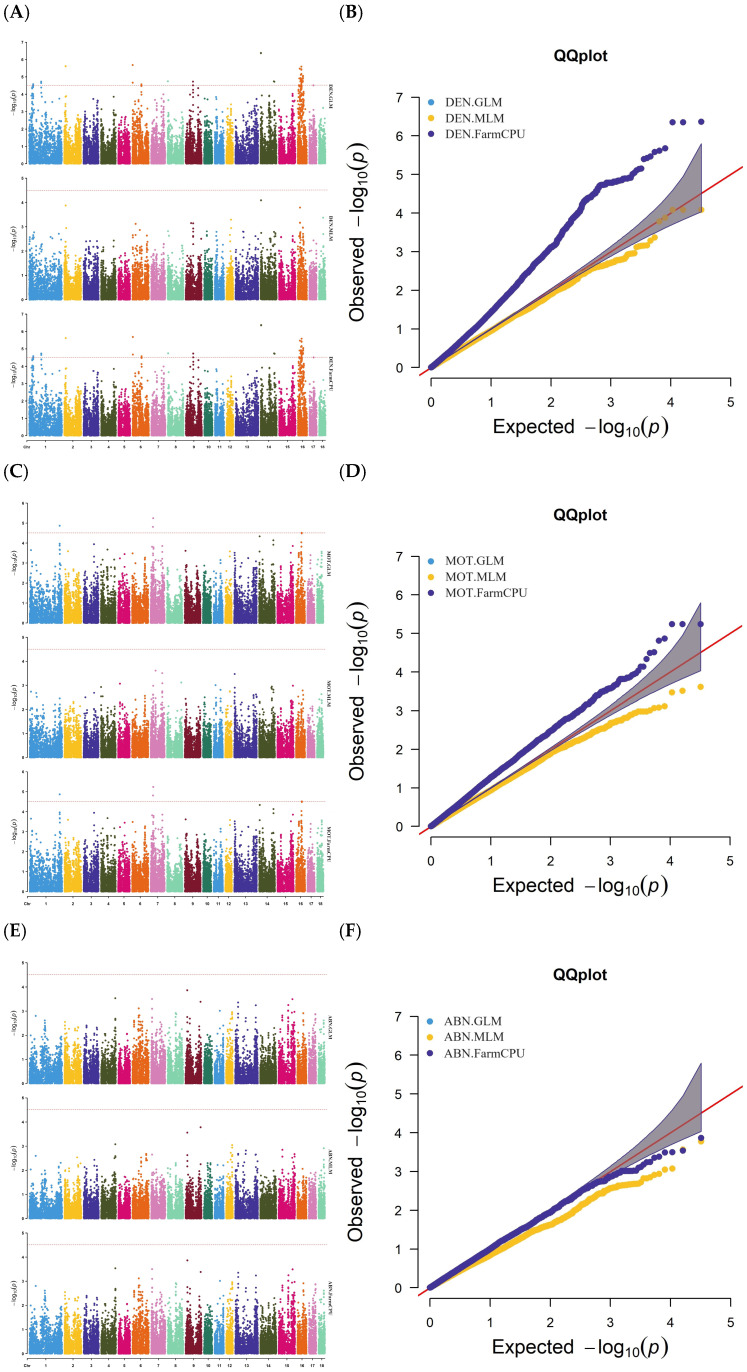
Manhattan and QQ plots obtained from the GWAS of DEN, MOT, ABN, MMP, AIR, and ROS traits in Duroc boars by the three models. The *x*-axis represents the chromosomes, and the *y*-axis represents the −log_10_(*p*-value). The dashed lines indicate the thresholds for semen traits in pigs after Bonferroni correction. The dashed line in Manhattan plots (**A**,**C**,**E**,**G**,**I**,**K**) indicates the thresholds for semen traits in pigs. Traits considered are semen density (**A**,**B**); sperm motility (**C**,**D**); abnormal sperm number (**E**,**F**); mitochondrial membrane potential (**G**,**H**); sperm acrosomal integrity rate (**I**,**J**); and reactive oxygen species level (**K**,**L**). Abbreviations: GLM = generalized linear models; MLM = mixed linear model; FarmCPU = fixed and random model circulating probability unification.

**Table 1 animals-15-00026-t001:** Descriptive statistics on quality traits of semen in Duroc boars.

Trait	N	Max	Min	Mean	SD	CV (%)
DEN (×10^9^/mL)	1169	13.83	1.08	6.42	2.31	35.98
MOT (%)	1162	99.00	79.00	90.63	3.70	4.08
ABN (%)	1169	20.00	1.00	6.32	4.12	65.19
AIR (%)	1097	86.34	29.31	57.86	9.50	16.42
MMP (%)	1055	92.37	48.01	79.55	8.36	10.51
ROS (%)	1012	32.54	0.10	6.75	6.14	90.96

DEN, semen density; MOT, sperm motility; ABN, abnormal sperm number; AIR, sperm acrosomal integrity rate; MMP, mitochondrial membrane potential; ROS, reactive oxygen species; N, number; Max, maximum; Min, minimum.

**Table 2 animals-15-00026-t002:** Estimated heritability of sperm quality traits by different animal models.

Trait	Models	${\sigma 2}_{a}^{2}$ (SE)	$\sigma_{e}^{2}$ (SE)	h^2^ (SE)
DEN	Mode1 1	1.58 (0.43)	3.74 (0.37)	0.30 (0.08)
	Mode1 2	1.56 (0.42)	3.75 (0.37)	0.29 (0.07)
MOT	Mode1 1	2.20 (0.84)	11.22 (0.85)	0.16 (0.06)
	Mode1 2	2.54 (0.87)	10.92 (0.85)	0.19 (0.06)
ABN	Mode1 1	1.47 (0.65)	13.25 (0.77)	0.10 (0.04)
	Mode1 2	1.84 (0.85)	12.86 (0.91)	0.13 (0.06)
AIR	Mode1 1	14.96 (5.71)	71.36 (5.69)	0.17 (0.06)
	Mode1 2	15.30 (5.69)	71.40 (5.67)	0.18 (0.06)
MMP	Mode1 1	8.06 (4.02)	57.09 (4.30)	0.12 (0.06)
	Mode1 2	7.17 (3.69)	57.81 (4.12)	0.11 (0.06)
ROS	Mode1 1	4.17 (2.35)	30.08 (2.45)	0.12 (0.07)
	Mode1 2	4.85 (2.37)	29.50 (2.43)	0.14 (0.07)

Abbreviations: $\sigma_{a}^{2}$ = Additive genetic variance; $\sigma_{e}^{2}$ = Residual variance; h^2^ = heritability Model 1 = multi-trait PBLUP animal model; Model 2 = single-trait PBLUP animal model. Standard errors are in parentheses. Heritability is narrowly defined as the ratio of the additive variance component to the total phenotypic variance component.

**Table 3 animals-15-00026-t003:** Estimates of genetic and phenotypic correlations of semen traits.

Trait	DEN	MOT	ABN	AIR	MMP	ROS
DEN	--	0.86 (0.13)	0.01 (0.30)	0.22 (0.22)	0.14 (0.27)	0.18 (0.25)
MOT	0.32 (0.03)	--	0.08 (0.28)	0.36 (0.23)	0.11 (0.29)	0.19 (0.28)
ABN	0.11 (0.03)	−0.12 (0.03)	--	0.28 (0.28)	0.15 (0.34)	0.00 (0.33)
AIR	0.11 (0.03)	0.17 (0.03)	0.09 (0.03)	--	0.93 (0.20)	−0.57 (0.28)
MMP	0.09 (0.03)	0.14 (0.03)	−0.01 (0.03)	0.38 (0.03)	--	−0.73 (0.31)
ROS	0.06 (0.03)	0.04 (0.03)	0.16 (0.03)	−0.06 (0.03)	−0.17 (0.03)	--

Note: Genetic correlations are shown in the upper right panel, phenotypic correlations are shown in the lower left panel, and standard deviations are in parentheses.

**Table 4 animals-15-00026-t004:** Important candidate genes affecting Duroc semen traits.

Trait	SNP	Gene Name	Gene Position	Biological Function
MOT	ALGA0039452	*GPX5*	7:22291986–22300727	Sperm antioxidant and vitality maintenance
AIR	WU_10.2_6_43634682	*AWN*	14:132240887–132248033	Sperm acrosome membrane structure and function
		*PSP-II*	14:132274241–132281728	Ditto
	MARC0041976	*CCDC62*	14:30028566–30066257	Ditto
MMP	MARC0029602	*TMEM65*	4:15263745–15352225	Involved in mitochondrial oxidative stress
	H3GA0039852	*SLC8B1*	14:38773407–38808265	Correlation of intracellular Ca^2+^ regulation
	MARC0016119	*TRPV4*	14:41125869–41169578	Ditto
	ALGA0077164	*UBE3B*	14:41383239–41442476	Ditto
ROS	WU_10.2_7_10722135	*SIRT5*	7:9951030–9968448	Involved in the regulation of antioxidant responses

## Data Availability

The original contributions presented in the study are included in the article/Appendix A, further inquiries can be directed to the corresponding author.

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
