# Peer review of "Genome-Wide Association Analysis of Boar Semen Traits Based on Computer-Assisted Semen Analysis and Flow Cytometry"

_animals, 2024, doi:10.3390/ani15010026_

Round 1

Reviewer 1 Report

Comments and Suggestions for Authors

Dear authors, congratulations on your study. Both the article and the supplementary materials reflect the meticulous effort invested in this work, recommending it for publication. The article addresses a current topic that could yield significant results in improving the pig farming sector. A few more objective remarks are provided below.

Lines 10-27 The simple summary needs to be more concise, with a maximum length of 200 words. This ensures the key points are communicated clearly and efficiently, making it easier for the audience to understand and retain the information.

Line 46  I suggest to also  include the keyword “CASA”

According to the instructions provided to the authors, references should be numbered in order of appearance and indicated by numerals in square brackets—for example, [1], [2,3], or [4–6]. Please revise all article in this direction.

In the introduction section, it is mandatory to include the methods for analyzing semen. Additionally, the practical aspects of using the CASA system in daily semen analysis should be mentioned, not only for boars but also for other species such as dogs, bulls, and rams (e.g., Effect of L-arginine and Eugenol on Ram Semen Kinematic Parameters and Post-Thawed Fertility Rate after Trans-cervical Artificial Insemination). The introduction should conclude with a sentence that clearly establishes the purpose of the study.

Line 95 replace the term “pig house” with pig barn

The rest of the text requires a full revision, as there are minor errors in some parts, such as missing spaces or extra spaces between certain words or sentences.

Author Response

Dear Reviewer,Thank you very much for your thorough and constructive feedback on our manuscript. We greatly appreciate the time and effort you have invested in reviewing our work and are pleased to hear that you recommend it for publication. We have taken your comments to heart and have made the following revisions to our manuscript:

Comments 1:Lines 10-27 The simple summary needs to be more concise, with a maximum length of 200 words. This ensures the key points are communicated clearly and efficiently, making it easier for the audience to understand and retain the information.

Response 1:We have revised the simple summary to be more concise, ensuring it is within the 200-word limit. Lines 10-22

Comments 2:Line 46  I suggest to also  include the keyword “CASA”

Response 2:As you suggested, we have included the keyword “CASA”.Lines 37

Comments 3:According to the instructions provided to the authors, references should be numbered in order of appearance and indicated by numerals in square brackets—for example, [1], [2,3], or [4–6]. Please revise all article in this direction.

Response 3:We have revised the entire article to ensure that all references are numbered in the order of appearance and indicated by numerals in square brackets, following the format you provided.Lines 388-524

Comments 4:In the introduction section, it is mandatory to include the methods for analyzing semen. Additionally, the practical aspects of using the CASA system in daily semen analysis should be mentioned, not only for boars but also for other species such as dogs, bulls, and rams (e.g., Effect of L-arginine and Eugenol on Ram Semen Kinematic Parameters and Post-Thawed Fertility Rate after Trans-cervical Artificial Insemination). The introduction should conclude with a sentence that clearly establishes the purpose of the study.

Response 4:Thank you for pointing this out. We agree with this comment.We have made the corresponding modifications according to your suggestions.Lines 55-58;Lines 73-74

Comments 5:Line 95 replace the term “pig house” with pig barn.

Response 5:We have replaced the term "pig house" with "pig barn" to maintain consistency and accuracy in our terminology.Lines 82-83

Comments 6:The rest of the text requires a full revision, as there are minor errors in some parts, such as missing spaces or extra spaces between certain words or sentences.

Response 6:We have conducted a full revision of the text, correcting minor errors such as missing or extra spaces between words or sentences to ensure the manuscript's readability and professionalism.

Reviewer 2 Report

Comments and Suggestions for Authors

Simple summary: looks good, no changed necessary

Abstract: authors don’t have to mention same abbreviations which they already did in simple summary

Introduction: I think introduction is the best place where authors can explain all abbreviations. In simple summary and abstract they can use whole phrase. Once explain abbreviations in introduction they don’t need to explain anymore.

Material and methods: In line 92 authors mention about “Animal Care and Use Committee of the Foshan University”, is this something affiliated with government? Or if there is any general body name you may wan to use. Like in USA they use IACUC.

Results: they looked good, no comments

Discussions: looks good, no changed necessary

Conclusion: no comments

Author Response

Dear Reviewer,Thank you very much for your recognition and support of our research.Your positive evaluation is a great encouragement to us. Based on your valuable feedback, we have made the following detailed revisions to our manuscript:

Comments 1:Abstract: authors don’t have to mention same abbreviations which they already did in simple summaryï¼›Introduction: I think introduction is the best place where authors can explain all abbreviations. In simple summary and abstract they can use whole phrase. Once explain abbreviations in introduction they don’t need to explain anymore.

Response 1: Thank you for pointing this out.According to your suggestion, abbreviations were used for simple summaries and abstracts, and only the abbreviations were explained in the introduction section.

Comments 2:Material and methods: In line 92 authors mention about “Animal Care and Use Committee of the Foshan University”, is this something affiliated with government? Or if there is any general body name you may wan to use. Like in USA they use IACUC.

Response 2:Animal Care and Use Committee established at Foshan University.The Animal Care and Use Committee is an independent non-governmental organization dedicated to promoting and advancing human protection and respect for animals through advocacy and legislation.

Reviewer 3 Report

Comments and Suggestions for Authors

Review comments/Report

In this manuscript the authors aim to identify genetic markers and candidate genes associated with key semen traits in Duroc boars, such as sperm motility, semen density, and mitochondrial membrane potential, using genome-wide association studies (GWAS). Their findings provide insights into the genetic basis of semen quality, which could enhance breeding strategies for improving reproductive performance in pig production. However, the presentation and expression of the content presented in their study is still lacking clarity in its expression. Therefore, Major revision is still required before publication of this manuscript in Animals. The section wise comments are listed as;

Abstract

Line 29-30: We conducted a genome-wide association study (GWAS) on various semen traits of Duroc boars... – The phrase "on various semen traits" could be more specific. What exactly are these traits? Sperm motility, semen density, etc. should be listed here in a clear, systematic way.

Line 38-39: Heritability estimates were 0.19, 0.29, 0.13, 0.18, 0.11, and 0.14 for MOT, DEN, ABN, MMP, AIR, and ROS, respectively. – These heritability values are fine but would benefit from a brief comment explaining what the values suggest (e.g., low or moderate heritability).

Line 42-44: Our findings provide insight into the genetic architecture of semen traits in Duroc boars." – The connection between these findings and their broader relevance to pig breeding could be emphasized more.

How to improve :Clarify semen traits: We conducted a genome-wide association study (GWAS) on various semen traits in Duroc boars, including sperm motility (MOT), semen density (DEN), the number of abnormal sperm (ABN), mitochondrial membrane potential (MMP), sperm acrosomal integrity rate (AIR), and levels of reactive oxygen species (ROS)." Include brief explanations for heritability estimates and their significance in the context of breeding.

Introduction

Line 51-54: A decline in semen quality is a major factor for the reduced lifespan of boars. – The statement is vague and lacks supporting evidence. How does semen quality directly impact boar lifespan? This could use some citation or more detailed explanation.

Line 56-58: Currently, boars are mainly selected based on growth, carcass, and reproductive traits, with less emphasis on the genetic parameters of semen quality. – This statement could be nuanced. While it’s true, there has been growing interest in semen quality, which could be expanded upon with more recent references.

Suggestions for Improvement: Provide clearer evidence or references for the statement that semen quality influences boar lifespan. Expand on the growing interest in semen quality and its role in breeding decisions, including recent trends or developments.

Material and Methods

• Line 93-94: "Between January 2022 and July 2022, semen samples were collected from 1,182 purebred Duroc boars..." – The year is quite specific, but it could be mentioned why this particular sampling period was chosen (e.g., seasonality, sampling consistency).

• Line 101-105: "CASA was used to measure MOT, DEN, and ABN... MMP was evaluated using Guava (HT 5.0) FCM..." – There is no explanation about the reliability or validation of these methods in the context of semen trait measurement.

• Line 105-106: "ROS levels were measured using Guava FCM and an ROS assay kit..." – ROS quantification methods could be further explained. What’s the sensitivity, and why was Guava FCM chosen?

Suggestions for Improvement:

• Provide justification for the sampling period (January–July 2022) and clarify if this timeline was optimal for semen quality assessment.

• Include citations or references for the validation of CASA and FCM as accurate methods for semen trait measurements.

• Mention the validation or any known limitations of the ROS assay kit and FCM methods used.

Results

• Line 207-209: "Phenotypic distribution is shown in Supplementary Fig. S1, demonstrating that all the traits were normally distributed." – This could be expanded. Were there any outliers or skewness present despite normal distribution? Were any transformations applied to the data?

• Table 1: "Descriptive of statistics on quality traits of semen in Duroc boars." – The table shows a wide variation in some traits, especially ROS with a CV of 90.96%. It would be helpful to explain if such a large variation is expected, and whether it might skew downstream analyses.

• Line 220-227: "Estimates of heritability for both DEN and MMP traits using the single-trait PBLUP model were lower than those from the multiple-trait model..." – This could benefit from some statistical clarification. Why were the heritability estimates lower in the single-trait model? Were there any significant statistical differences?

Suggestions for Improvement:

• Expand on the phenotypic distribution findings and comment on any transformations applied. • Provide some context or justification for the high CV observed in ROS levels, especially in relation to the breeding context.

• Clarify why the single-trait model gave lower estimates for DEN and MMP, and explain the implications of this in the context of trait selection.

Discussion

• Line 311-319: "The heritability estimates for DEN, MOT, and ABN are consistent with those reported by Hong et al., although the estimates for MMP, AIR, and ROS levels are novel..." – The discussion would benefit from a more in-depth comparison with previous studies on semen traits, explaining how these findings expand the current understanding of genetic selection.

• Line 335-345: "Genetic correlations were assessed using a multi-trait PBLUP model..." – While genetic correlations are discussed, it could be expanded upon how these correlations may guide breeding practices. For example, a positive correlation between DEN and MOT suggests selection for one could improve the other.

Suggestions for Improvement:

• Compare the results of heritability estimates with more studies and discuss the implications of novel findings (such as for MMP, AIR, and ROS).

• Further explain how genetic and phenotypic correlations might influence breeding decisions, particularly in terms of indirect selection.

Conclusion

• Line 377-384: "These genes are involved in crucial functions, such as sperm antioxidation, acrosome membrane structure and function, mitochondrial oxidative stress response..." – While the genes are listed with their functions, this section could further discuss how these findings can directly influence breeding practices or how these traits impact reproductive performance.

• The conclusion briefly mentions "marker-assisted genomic selection," but does not explain how these results will be incorporated into actual breeding programs.

Suggestions for Improvement:

• Provide more explicit details about the potential for these findings to influence breeding programs. For example, how would candidate genes like GPX5 for MOT or SIRT5 for ROS be utilized in breeding practices?

• Emphasize the broader implications of identifying these genes for improving boar semen quality in a commercial setting.

Tables and Figures

• Table 1: The units for DEN are not specified in the column header. This could cause confusion, as semen density is typically measured in units like 10^9/mL. Clarify the units for all traits.

• Table 2: The table showing heritability estimates would benefit from clarification regarding the specific models used, especially when there are small differences in heritability between models. An explanation of why these differences exist could help interpret the results.

• Figure 1: The Manhattan plot legends (e.g., "Dashed line indicates the thresholds...") are helpful, but additional clarity about the significance level used (e.g., Bonferroni correction or FDR) could aid interpretation.

Suggestions for Improvement:

• Clearly define units in Table 1 and consider adding standard deviations or confidence intervals for all reported traits.

• Provide a clearer explanation in the figure legends, particularly regarding significance thresholds for the SNPs in the Manhattan plots.

Author Response

Dear Reviewer,Thank you for your meticulous reading of our manuscript and for your valuable comments. Your professional suggestions are extremely precious to us, as they not only help us improve the quality of our research but also enhance the scientific rigor and persuasiveness of our study. We have carefully considered each of your recommendations and have made corresponding revisions and improvements to the article. Below are the modifications and responses we have made in response to your specific suggestions:

Comments 1: Line 29-30: We conducted a genome-wide association study (GWAS) on various semen traits of Duroc boars... – The phrase "on various semen traits" could be more specific. What exactly are these traits? Sperm motility, semen density, etc. should be listed here in a clear, systematic way.

Response 1: Line 24-25: We conducted a GWAS on various semen traits of Duroc boars, including MOT, DEN, ABN, MMP, AIR, and ROS levels. 

Comments 2:Line 38-39: Heritability estimates were 0.19, 0.29, 0.13, 0.18, 0.11, and 0.14 for MOT, DEN, ABN, MMP, AIR, and ROS, respectively. – These heritability values are fine but would benefit from a brief comment explaining what the values suggest (e.g., low or moderate heritability). 

Response 2: Line 30: All semen traits exhibited low heritability except ABN, which demonstrated medium heritability. 

Comments 3:Line 42-44: Our findings provide insight into the genetic architecture of semen traits in Duroc boars." – The connection between these findings and their broader relevance to pig breeding could be emphasized more. 

Response 3: Line 33-36: We agree that it is important to emphasize the implications of our research for the field of pig breeding. 

Comments 4:Line 51-54: A decline in semen quality is a major factor for the reduced lifespan of boars. – The statement is vague and lacks supporting evidence. How does semen quality directly impact boar lifespan? This could use some citation or more detailed explanation. 

Response 4: Line 44-46: The decline in semen quality is one of the main factors leading to a shortened lifespan of boars, as boars with poor semen quality will be elimi nated.

Comments 5:Line 56-58: Currently, boars are mainly selected based on growth, carcass, and reproductive traits, with less emphasis on the genetic parameters of semen quality. – This statement could be nuanced. While it’s true, there has been growing interest in semen quality, which could be expanded upon with more recent references. 

Response 5: Line 48-57: We have expanded based on the latest reference materials.

Comments 6: Line 93-94: "Between January 2022 and July 2022, semen samples were collected from 1,182 purebred Duroc boars..." – The year is quite specific, but it could be mentioned why this particular sampling period was chosen (e.g., seasonality, sampling consistency). 

Response 6: Line 92-95: Actually, we originally intended to collect data for one year, but due to some special reasons, we were not successful.

Comments 7:Line 101-105: "CASA was used to measure MOT, DEN, and ABN... MMP was evaluated using Guava (HT 5.0) FCM..." – There is no explanation about the reliability or validation of these methods in the context of semen trait measurement. 

Response 7: Line 100-105: We have introduced references for each detection method.

Comments 8: Line 105-106: "ROS levels were measured using Guava FCM and an ROS assay kit..." – ROS quantification methods could be further explained. What’s the sensitivity, and why was Guava FCM chosen?

Response 8: Line 105-108: Flow cytometry has the advantages of being fast, objective, multi index, and high-throughput, and the technology for measuring reactive oxygen species levels using flow cytometry is currently quite mature.

Comments 9: Line 207-209: "Phenotypic distribution is shown in Supplementary Fig. S1, demonstrating that all the traits were normally distributed." – This could be expanded. Were there any outliers or skewness present despite normal distribution? Were any transformations applied to the data? 

Response 9: Lines 109-111 provide a detailed explanation of the quality control standards for phenotype data.

Comments 10: Table 1: "Descriptive of statistics on quality traits of semen in Duroc boars." – The table shows a wide variation in some traits, especially ROS with a CV of 90.96%. It would be helpful to explain if such a large variation is expected, and whether it might skew downstream analyses.

Response 10: Lines 211-214: We have added the reason for the large coefficient of variation in reactive oxygen species levels. The factors contributing to the increase in ROS are multifaceted. In addition to heat stress, elements such as the types and quantities of microorganisms present in semen, non-standardized semen collection procedures, and inadequate detection methods may also play a significant role in this phenomenon. 

Comments 11: Line 220-227: "Estimates of heritability for both DEN and MMP traits using the single-trait PBLUP model were lower than those from the multiple-trait model..." – This could benefit from some statistical clarification. Why were the heritability estimates lower in the single-trait model? Were there any significant statistical differences? 

Response 11:  Line 227-229: The multiple-trait model may yield higher estimates of heritability compared to the single-trait model by leveraging the genetic correlations among traits and more effectively decomposing additive and non-additive genetic effects. 

Comments 12: Line 311-319: "The heritability estimates for DEN, MOT, and ABN are consistent with those reported by Hong et al., although the estimates for MMP, AIR, and ROS levels are novel..." – The discussion would benefit from a more in-depth comparison with previous studies on semen traits, explaining how these findings expand the current understanding of genetic selection. 

Response 12: Line 327-348:We have listed some previous research results, but there are significant differences in the genetic backgrounds of different pig populations, and the estimated semen genetic parameters reported in different reports cannot be directly compared and analyzed. 

Comments 13: Line 335-345: "Genetic correlations were assessed using a multi-trait PBLUP model..." – While genetic correlations are discussed, it could be expanded upon how these correlations may guide breeding practices. For example, a positive correlation between DEN and MOT suggests selection for one could improve the other.

Response 13: Line 360-362: There is a positive correlation between DEN and MOT. In practical production, it is common to select semen with higher density to improve the success rate of breeding.

Comments 14: Line 377-384: "These genes are involved in crucial functions, such as sperm antioxidation, acrosome membrane structure and function, mitochondrial oxidative stress response..." – While the genes are listed with their functions, this section could further discuss how these findings can directly influence breeding practices or how these traits impact reproductive performance.

Response 14: Line 399-405: We have carefully considered your suggestion and added relevant content to the manuscript.

Comments 15: Table 1: The units for DEN are not specified in the column header. This could cause confusion, as semen density is typically measured in units like 10^9/mL. Clarify the units for all traits.

Response 15:  Line 219: Table 1, we indicate the units for each trait.

Comments 16: Figure 1: The Manhattan plot legends (e.g., "Dashed line indicates the thresholds...") are helpful, but additional clarity about the significance level used (e.g., Bonferroni correction or FDR) could aid interpretation.

Response 16: Thank you for pointing this out. We agree with this comment. The dashed lines indicate the thresholds for semen traits in pigs after Bonferroni correction. Line 296-297

Round 2

Reviewer 3 Report

Comments and Suggestions for Authors

The authors have adequately addressed all major and minor comments raised by me. The revisions have significantly improved the clarity, scientific rigor, and presentation of the manuscript. Based on the thorough revisions and the overall quality of the manuscript, I recommend this paper for acceptance.